# Solar-Induced Photocatalytic Degradation of Reactive Red and Turquoise Dyes Using a Titanium Oxide/Xanthan Gum Composite

**Abeer I. Alwared [1], Noor A. Mohammed [1], Tariq J. Al-Musawi [2,\*] and Ahmed A. Mohammed [1]**

[1] Department of Environmental Engineering, College of Engineering, University of Baghdad, Baghdad 10071, Iraq

[2] Building and Construction Techniques Engineering Department, Al-Mustaqbal University, Hillah 51001, Iraq

\* Correspondence: tariq.jwad@mustaqbal-college.edu.iq

**Abstract:** The present study explores the solar-induced photocatalytic degradation of reactive red (RR) and reactive turquoise (RT) dyes in a single system using $TiO_2$ immobilized in xanthan gum ($TiO_2/XG$), synthesized using the sol–gel dip-coating technique for direct precipitation. SEM-EDX, XRD, FTIR, and UV–Vis were used to assess the characteristics of the resulting catalyst. Moreover, the effects of different operating parameters, specifically pH, dye concentration, $TiO_2/XG$ concentration, $H_2O_2$ concentration, and contact time, were also investigated in a batch photocatalytic reactor. The immobilized $TiO_2/XG$ catalyst showed a slight adsorption degradation efficiency and then improved the RR and RT dye degradation activity (92.5 and 90.8% in 120 min) under solar light with a remarkable Langmuir–Hinshelwood pseudo-first-order degradation rate of 0.0183 and 0.0151 $min^{-1}$, respectively, under optimum conditions of pH 5, dye concentration of 25 mg/L, $TiO_2/XG$ concentration of 25 mg/L, $H_2O_2$ concentration of 400 mg/L, and reaction time of 120 min. The improved photocatalytic ability was ascribed to the impact of $TiO_2/XG$ nanoparticles with a high surface area, and lower band gap energy. Solar light energy has significant potential for addressing energy deficit and water pollution concerns.

**Keywords:** $TiO_2$; xanthan gums; immobilization; photocatalyst batch process; red and turquoise dye; pseudo-first-order kinetics

## 1. Introduction

Energy consumption and environmental pollution have always been difficult problems worldwide [1,2] Water pollution has attracted significant attention as a result of the discharge of a variety of unwanted pollutants into the environment. These can come from a variety of sources, such as industries, municipalities, and agricultural practices [3,4].

The textile industry has been shown to be a particularly high source of pollutants, generating a significant amount of wastewater daily. Its affluent often contains a high concentration of organic compounds, dyes, and other nutrients, which, if not treated properly, can lead to major environmental pollution [5], such as acarbose [6], and rhodamine-610 [7]. Even at concentrations less than 1 mg/L, the presence of traces of dye in water is very apparent and influences the aesthetic value of lakes, rivers, and other water bodies, impacting their water transparency and gas solubility. However, dyes are more difficult to handle than other contaminants because of their artificial origin and complex aromatic structures [8]. Eliminating colors from wastewater that has been significantly contaminated can be accomplished via photolysis, adsorption, electrochemical or chemical precipitation, membrane approaches, chemical oxidation and reduction, or any combinations thereof. However, many of these methods have significant drawbacks, such as high costs, the need for complex conditions for the reaction, or the fact that they do little more than transform the contaminants from liquids to solids, as is the case with most adsorption techniques [9].

Emerging technologies, such as advanced oxidation processes (AOPs), are becoming more enticing as alternatives due to their broad range of applications in the elimination of toxic pollutants [10,11]. Among different AOP techniques, photocatalytic oxidations have been highly promoted as inexpensive approaches with good removal effectiveness compared to other AOPs [11,12]. The mechanism of photocatalysis is primarily described by the semiconductors' capability to generate charge carriers under light irradiation, followed by the generation of free radicals such as $OH^{-\prime}$ which leads to further reactions, eventually forming $CO_2$ and $H_2O$ [13,14]. The technology of photocatalysis has, in particular, sparked tremendous interest due to its ability to capture solar energy utilizing semiconductor materials that function as catalysts. Catalysts can aid in the resolution of the environmental challenges associated with water contamination, and these semiconductor materials are harmless and efficient [15]. $TiO_2$ has received a lot of attention among semiconductors because of its potential use in the breakdown of a wide range of environmental pollutants in both gaseous and liquid phases [16]. Numerous researchers have worked to improve the catalytic activity of $TiO_2$ by combining it with magnetic nanoparticles [17] or doping it with non-metallic elements [18–20]. On the other hand, the use of recycled plastic materials as a support or to immobilize photocatalyst powders has emerged as a new trend. This facilitates the removal of the photocatalytic material from the decontaminated water after the elimination of the contaminant using simple methods [13]. Activated carbon, silica, zeolites, alumina, carbon nanotubes, and cellulosic fibers produced from chestnut have been utilized to modify the surface or structure of $TiO_2$ [21,22]. Meanwhile, sunlight-induced photodegradation has recently emerged as a cutting-edge, environmentally friendly, and promising approach to wastewater treatment [23]. Xanthan gum (XG) is a naturally occurring polysaccharide consisting of repeating pentasaccharide units made up of two units of mannose, two units of glucose, and one unit of glucuronic acid (molar ratio of 2.8:2:2). The chemical structure of XG is similar to that of cellulose; it has no sensitizing characteristics, and it does not trigger allergic reactions in the skin or eyes. The characteristics of XG are its high viscosity, water solubility, heat stability, emulsion stabilization, and food component compatibility. In addition, it is a non-toxic material, which is especially important for water treatment systems. The Food and Drug Administration of the United States of America has authorized it as a food additive with no further restrictions, with pharmaceuticals, cosmetics, petroleum, and agriculture being its four main industrial areas of application [24]. This study aims to develop $TiO_2/XG$ and use it for removing reactive red (RR) and reactive turquoise (RT) dyes in their single system from an aqueous solution. The performance of the $TiO_2/XG$ is shown by studying the adsorbents and solar-induced photocatalysts and assessing functional groups, inner structure, constituent elements, and morphology of the $TiO_2/XG$ through analysis by corresponding techniques.

## 2. Materials and Methods

### 2.1. Materials

The used dyes (RR and RT) were sourced from AL-Kadhimiya Textile Company, Department of Dying and Printing, and were delivered from Thomas Baker, India. The relevant chemical and physical characteristics are given in Table 1 and Figure 1. XG, in the form of a yellow-white powder and with a 1.6 specific gravity, was sourced from Barazan D Plus and was purchased from Central Drug House, Solan, India. Finally, titanium tetra-isopropoxide (TTIP) (97%), hydrogen peroxide, and acetic acid were supplied from Merck.

**Table 1.** Properties of dyes used in this study.

| Item | Reactive Turquoise | Reactive Red |
|---|---|---|
| Trade name | Blue 77 | Red 3B |
| Origin | China | China |
| Phase | Solid/Powder Package 5 kg | Solid/Powder Package 5 kg |
| Wavelength (nm) | 665 | 540 |
| Solubility g/L | 100–300 | 100 |
| Purity | 99% | 99% |
| CAS | 12236-86-1 | 70210-20-7 |
| pH(1% aqueous solution) | $7 \pm 0.5$ | $7.2 \pm 0.3$ |
| Molecular formula | $C_{19}H_{10}Cl_2N_6Na_2O_7S_2$ | $C_{41}H_{25}ClCuN_{14}Na_4O_{14}S_5$ |
| Molecular weight | 615.3 | 1289.0 |

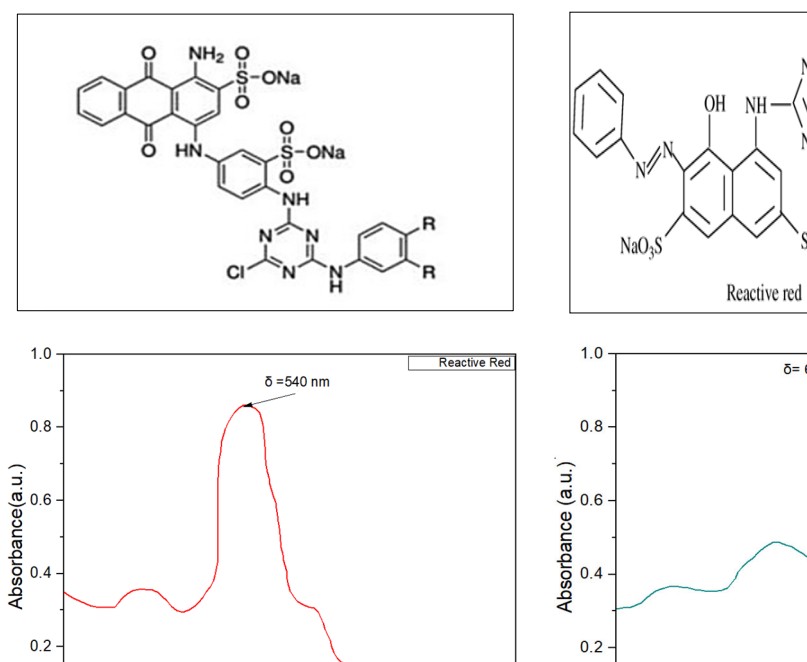

**Figure 1.** The dyes' chemical structure. (**a**) Reactive turquoise; (**b**) reactive red.

### 2.2. Fabrication of TiO$_2$/XG

TiO$_2$ was directly precipitated onto XG using the sol–gel dip-coating process (TiO$_2$/XG). First, 30 mL of the TTIP solution, 3.5 mL of the acetic acid, and 5 mL of the distilled water were stirred for 60 min. To this was added 0.2 gm of XG, and a shaker incubator was used to stir the mixture for 30 min at 350 rpm. Finally, the coated XG was dried in an oven at 50 °C for 30 min. The sample synthesis was then gathered and placed in a muffle furnace at 200 °C for 2 h for calcination. Lastly, it was completely cleansed with distilled water and dried for 30 min in a hot air oven at 50 °C [25]. A second coating cycle was then used to increase the thickness of the immobilized TiO$_2$ film (Figure 2). The TiO$_2$/XG was then washed with distilled water to eliminate any remaining disconnected TiO$_2$ particles, and the material was then saved for later usage.

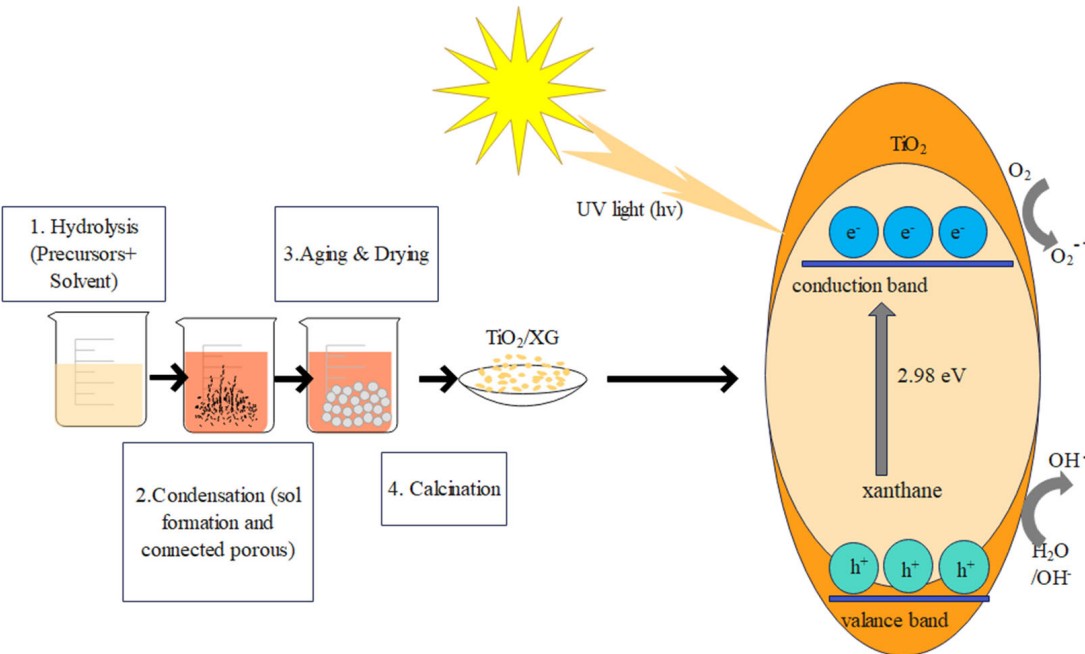

**Figure 2.** Schematic of the $TiO_2$/XG preparation.

### 2.3. Characterization of the $TiO_2$/XG Nanocomposite

The optical absorption of the nanoparticles sample was observed in the UV–Visible spectrum taken by UV–vis spectrometer (UV–Vis Spectrophotometer Perkin-Elmer 55, Waltham, MA, USA). The spectrum was recorded in the wavelength range of 200–600 nm. The optical band gap was measured using a UV–Vis spectrometer (UV-1800, SHIMADZU, Kyoto, Japan) to assess the $TiO_2$/XG nanocomposite's semiconductor nature. Subsequently, Tauc's plot double beam equation (Equation (1)) was used to estimate the band gap as appropriate for 200–800 nm wavelengths [26].

$$(\alpha h v)^n = A(h v - Eg) \tag{1}$$

where $\alpha$ is the coefficient of absorption, $A$ is a constant with different values for different transitions, e.g., the optical energy band gap value, $h$ is Planks constant ($6.626 \times 10^{-34}$ Joules sec), $h v$ is the light frequency, and $n$ is a constant number that depends on the transition characteristics and can be $1/2$, 2, $3/2$ and 3 for, respectively, allowed direct, allowed indirect, forbidden direct, and forbidden indirect transitions.

The various functional groups present on the surface of the $TiO_2$ sample were identified from the Fourier-transform infrared (FTIR) spectrum. The spectrum was recorded over the wavenumber range of 4000–500 $cm^{-1}$, recorded by an FTIR spectrometer (FT-NIR Spectrometer L1390022, PerkinElmer, Waltham, MA, USA). The topography of the nanoparticles sample was observed from the images taken by an emission scanning electron microscope (SEM TESCAN MIRA3, FRENCH, Fuveau, France). The structural properties, crystallite size, and interplanar spacing between the crystallographic planes of the nanoparticles were determined by X-ray diffraction (XRD) patterns obtained by an X-ray diffractometer (XRD PHILIPS XPERT, HOLLAND, Amsterdam, The Netherlands) and handled by X'Pert HighScore Plus software (version 2.0). The diffraction patterns were recorded in the 2θ range of 10°–80°. The operation voltage was 10 kV.

### 2.4. Experimental Procedure and Analysis

In a batch-mode reactor, two types of dye (RR and RT) were used to evaluate the process of adsorption and the photocatalytic characteristics of the produced materials (Figure 3). The reactor was made of Pyrex glass (1 L) and had a mirror at the bottom that

served as a reflector [27]. The solutions were exposed to direct solar radiation between 11 a.m. and 3 p.m. on consecutive sunny days in June and July 2022. Solar radiation was measured using an SPM-1116SD (LUTRON ELECTRONIC, Coopersburg, PA, USA), whereby the device was directed toward the sun to measure the solar radiation reaching the reactor. UV measurements were obtained for the range between 300 and 400 nm, with an average solar UV power of 150–850 W/m². Different concentrations of the dye solutions were prepared and adjusted to the desired pH using a dilute solution of 0.1 M HCl or NaOH (employing a pH meter model INOLAB 72, WTW Co., Weilheim, Germany). After that, the $TiO_2$/XG nanoparticles were added to the solution at the desired concentrations (15, 25, 50, and 75) mg/L. The suspension was agitated at 200 rpm (type MSH-300N, BOECO, Hamburg, Germany). The solution was kept in the dark under stirring for 30 min in order to reach the primary adsorption equilibrium between the dyes and the $TiO_2$/XG nanocomposite [28]. Consequently, the solutions were exposed to sunlight under constant stirring; $H_2O_2$ in various amounts (200, 400, and 600 mg/L) was added to start the reaction. Thereafter, 10 mL was regularly extracted and centrifuged at 200 rpm for 15 min to obtain the catalyst. The dye concentrations in the respective samples were assessed using a spectrophotometer (UV–Vis Spectrophotometer Perkin-Elmer 55 OSE) with a maximum of 540 and 665 nm for the RR and RT dyes, respectively. Finally, the percentage of dye elimination was estimated using Equation (2):

$$\text{dye elimination percentage} = \frac{\text{Initial dye conc.} - \text{Residual dye conc.}}{\text{Initial conc.}} \times 100 \qquad (2)$$

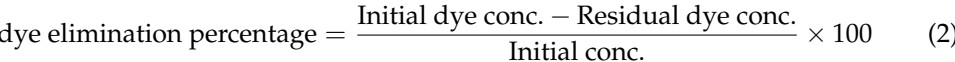

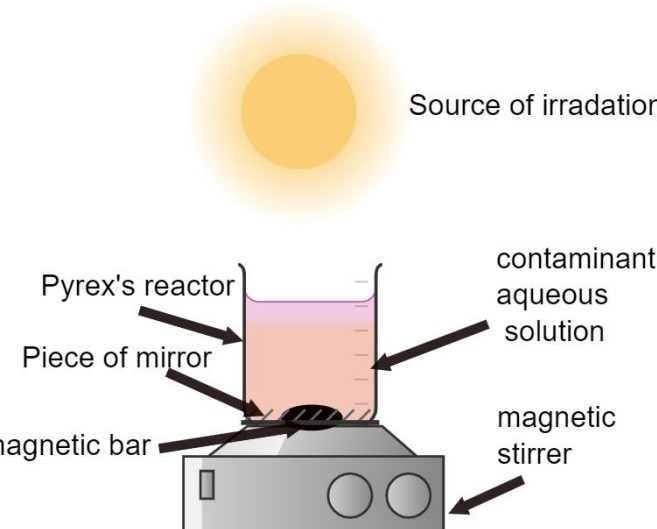

**Figure 3.** Schematic representation of the solar-powered photo-oxidation process.

## 3. Results and Discussion

### *3.1. TiO₂/XG Characteristics*

3.1.1. Optical Properties UV–Vis Spectroscopy

The $TiO_2$ crystal formations included rutile, anatase, and brookite. Photocatalytic studies have mainly concentrated on the rutile and anatase phases. Rutile is the most thermodynamically stable crystal structure of any $TiO_2$ phase, whereas anatase is metastable [29]. The wide range of band gap (3.2 ev) of anatase $TiO_2$ gives excellent photocatalytic activity by using UV radiation. In contrast, in the visible region of the solar spectrum, it presents inadequate efficiency [30,31]. As a result, the photocatalytic application of $TiO_2$ on an industrial scale has been limited because only 4–5% of the solar spectrum corresponds to UV photons. To this end, much effort has been expended to activate $TiO_2$ in the presence of visible and solar light for application purposes [31].

Figure 4a shows the UV–Vis spectrum (DRS) of the $TiO_2$/XG composite in the 200 to 800 nm range, which was carried out for the photo properties perception of $TiO_2$/XG composite nanoparticles. The main absorption bands at 365 nm were in the variety of 200 to 600 nm. Furthermore, the optical band gap results obtained by Tauc relation are shown graphically in Figure 4b using Origin Pro 8 software (Origin Lab, Northampton, MA, USA). The direct optical band gap of the $TiO_2$/XG nanocomposite was visible at 2.98 ev; due to the structural changes and greater light absorption, and the indirect band gap energy was 2.04 ev.

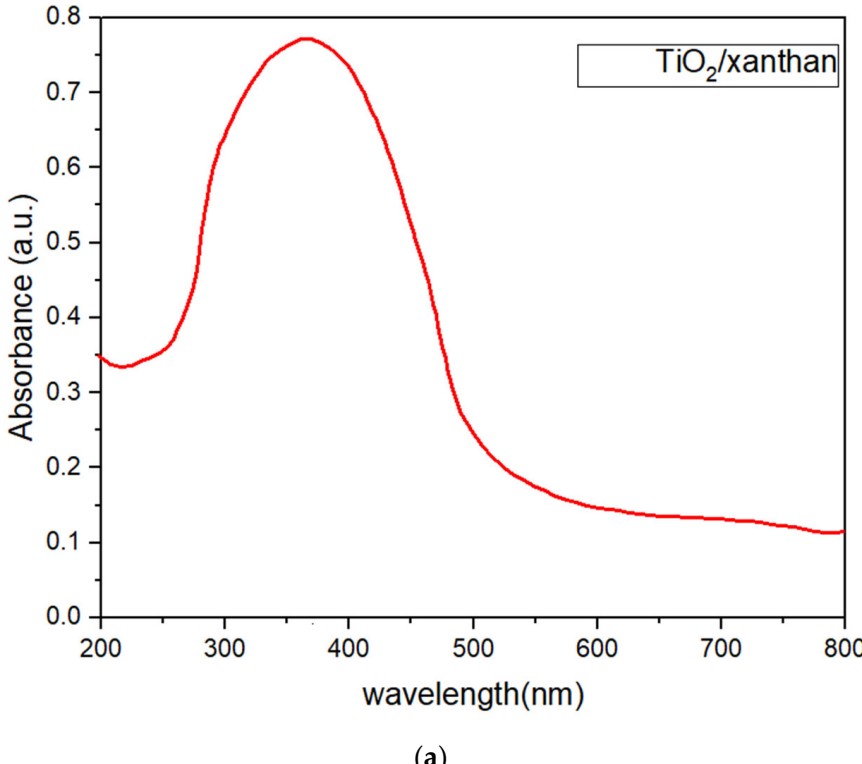

**(a)**

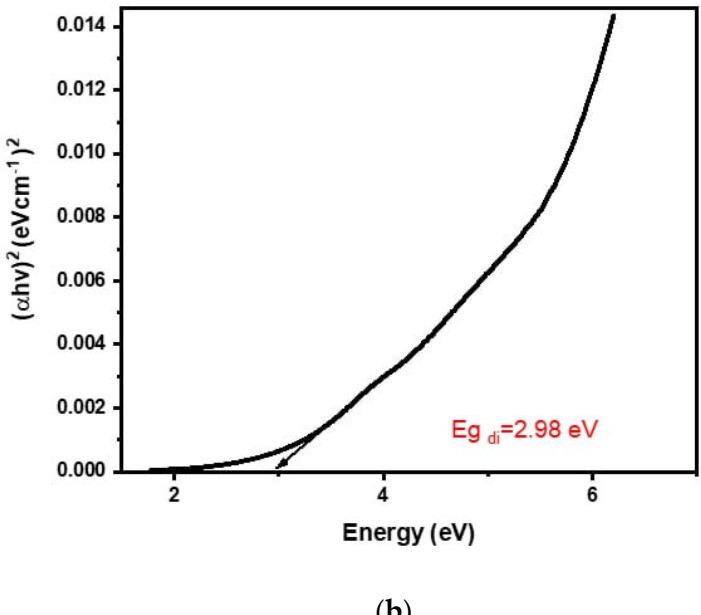

**(b)**

**Figure 4.** *Cont.*

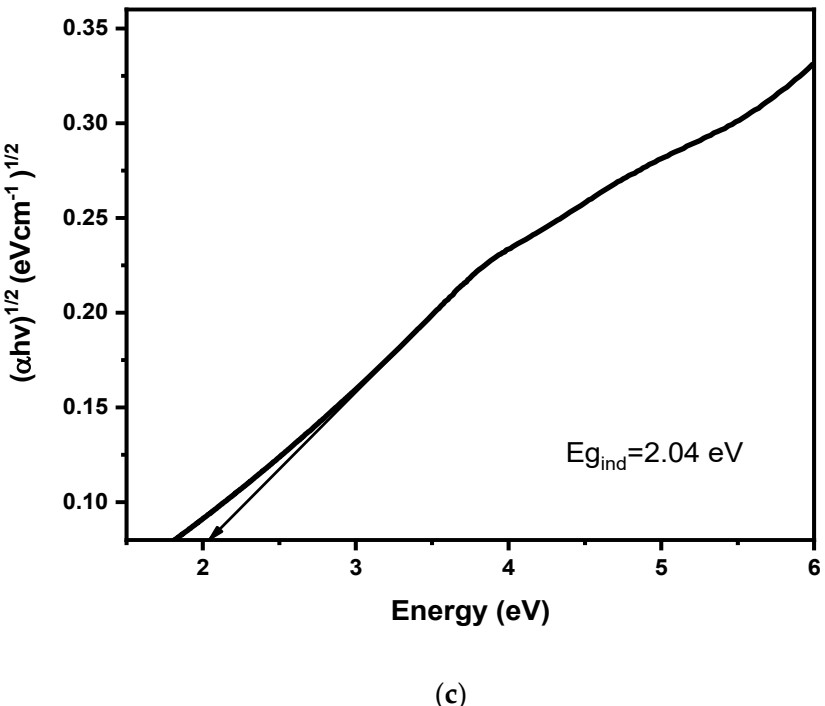

(**c**)

**Figure 4.** The band gap energy for $TiO_2$/XG. (**a**) Absorption spectra for $TiO_2$/XG nanoparticles, (**b**) direct band gap, (**c**) indirect band gap.

3.1.2. FTIR

The $TiO_2$/XG was investigated before and after the dye degradation reaction using FTIR analysis from 4000 to 500 cm$^{-1}$. Figure 5 depicts the FTIR spectra and their results. The FTIR spectrum in Figure 5 (black line) represents the peaks obtained for $TiO_2$/XG before being used in the degradation of dyes. It exhibits a prominent broad at 3419.17 cm$^{-1}$, signifying the stretching vibrations for the -OH groups of XG [32]. In addition, the peak detected at 2361.7 cm$^{-1}$ is linked to C=C conjugated and C≡C [33]. The peak at 1634.3 cm$^{-1}$ is linked to titanium carboxylate, which gives rise to the CH$_3$ stretching frequency peaks at 2921.3 cm$^{-1}$. The peaks detected between 500 and 800 cm$^{-1}$ were most likely caused by Ti-O stretching bands.

After dye degradation (red and blue curves for the solution of RR and RT dye in Figure 3, respectively), the intensity of some peaks shifted to new values due to the reaction between the dye molecules and the catalyst active groups [34]. Thus, there was a change in the peak intensity at 3419.7 cm$^{-1}$ compared to 3444.2 and 3427.8 cm$^{-1}$, which resulted from the interaction between the dyes with the -OH groups of XG in the $TiO_2$/XG nanocomposite, and at 2921.3 cm$^{-1}$ compared to 2920.1 and 2916.8 cm$^{-1}$, due to interaction between the groups in dyes with titanium carboxylate in addition to the peaks at 1634.3 cm$^{-1}$ compared to 1635.4 to 1633.3 cm$^{-1}$, shifting the peak at 668.2 cm$^{-1}$ to 560.2 and 521.6 cm$^{-1}$ for the RR and RT dyes, respectively. In addition, the results shown in this figure present the absence of the peak at 2381.7 cm$^{-1}$.

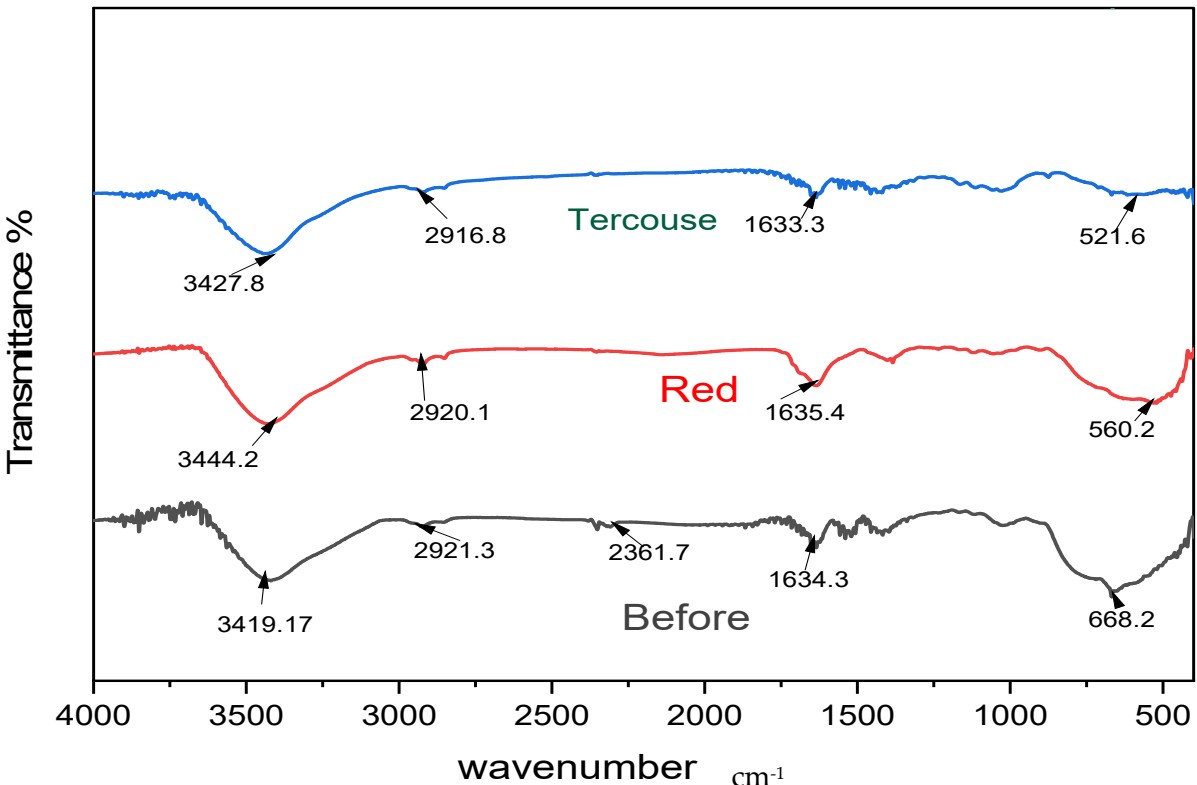

**Figure 5.** FTIR analysis for TiO$_2$/XG.

### 3.1.3. Scanning Electron Microscopy

Scanning electron microscopy (SEM) analysis (ZEISS model, Sigma V PEDS, and Oxford Instruments for mapping) was used to examine the morphology of TiO$_2$/XG, including particle size, shape, and surface properties, before and after use. As shown in Figure 6, there is some agglomeration of TiO$_2$/XG prior to treatment, which appears to be a cluster of irregularly shaped particles with a size of 24–40 nm. The images further show that after treatment (Figure 6b,c), the RR and RT dye adsorption completely covered the pores and surface.

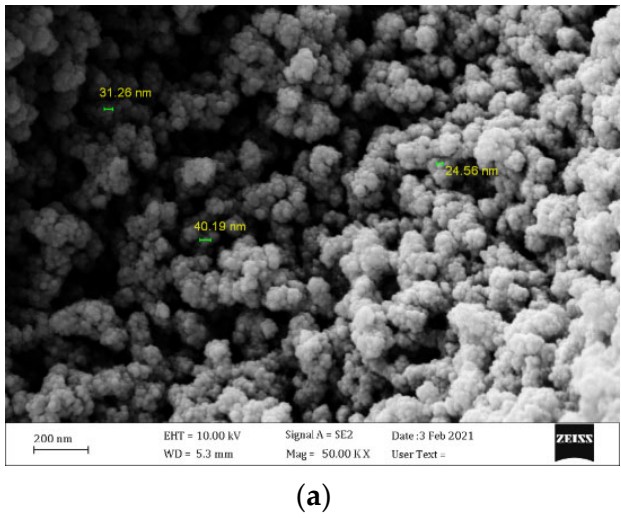

(**a**)

**Figure 6.** *Cont.*

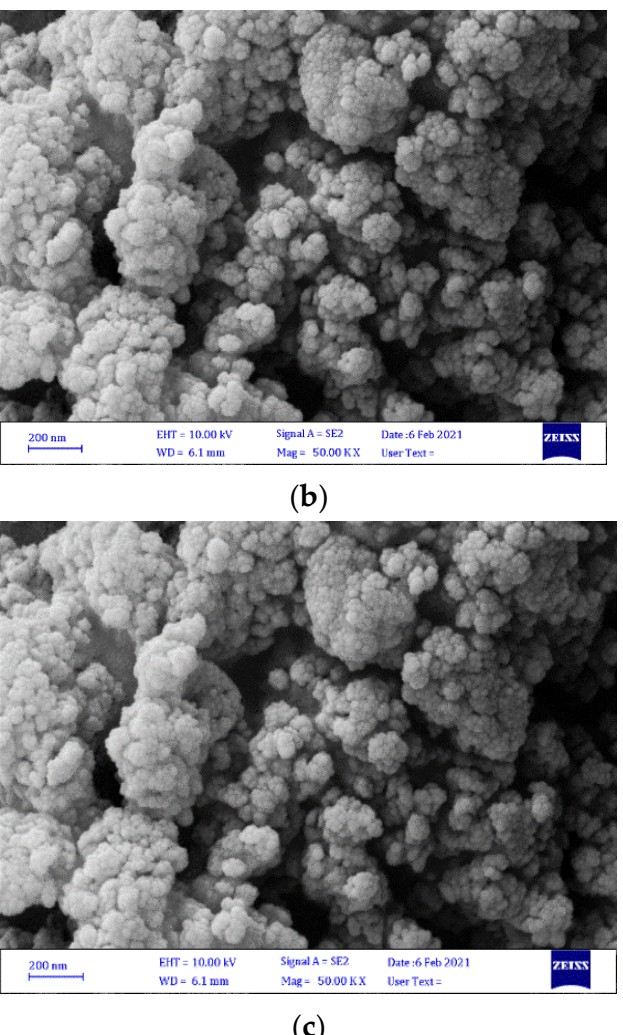

**Figure 6.** SEM analysis for TiO$_2$/XG. (**a**) TiO$_2$/XG (before treatment), (**b**) TiO$_2$/XG (RR dye treatment), (**c**) TiO$_2$/XG (RT dye treatment).

### 3.1.4. X-ray Diffraction

The X-ray diffraction pattern (XRD) is one of the most valuable scientific instruments for determining the crystalline nature of synthetic materials. The test was carried out using an XRD device (Siemens model D500, Munich, Germany). Figure 7 shows the XRD pattern for TiO$_2$/XG, visible at an angle from $10° < 2\theta < 80°$. The strongest peak was at $2\theta = 25.28°$, corresponding to TiO$_2$. Other peaks were seen at about $37°$, $48°$, $52°$, $63°$, $69°$, and $75°$, which were related to the (004), (200), (211), (213), (116), and (215) planes of the anatase due to the presence of XG in the XRD pattern of TiO$_2$/XG. After being used in the degradation of dyes, the small crystalline peaks disappeared, i.e., the loaded structures of TiO$_2$/XG became entirely amorphous.

The average crystal size for TiO$_2$/XG was 36.35 nm, calculated using Scherrer's equation,

$$D = \frac{K\lambda}{\beta cos\theta} \tag{3}$$

Where, $D$ represents crystallite size; $K$, Scherrer constant ($K = 0.94$); $\lambda$, the wavelength of X-ray radiation ($\lambda = 0.15418$ nm); $\beta$, full width at half maximum (FWHM) of the diffraction peak to the corresponding crystallographic plane of anatase; and $\theta$, the angle of the X-ray diffraction peak [35].

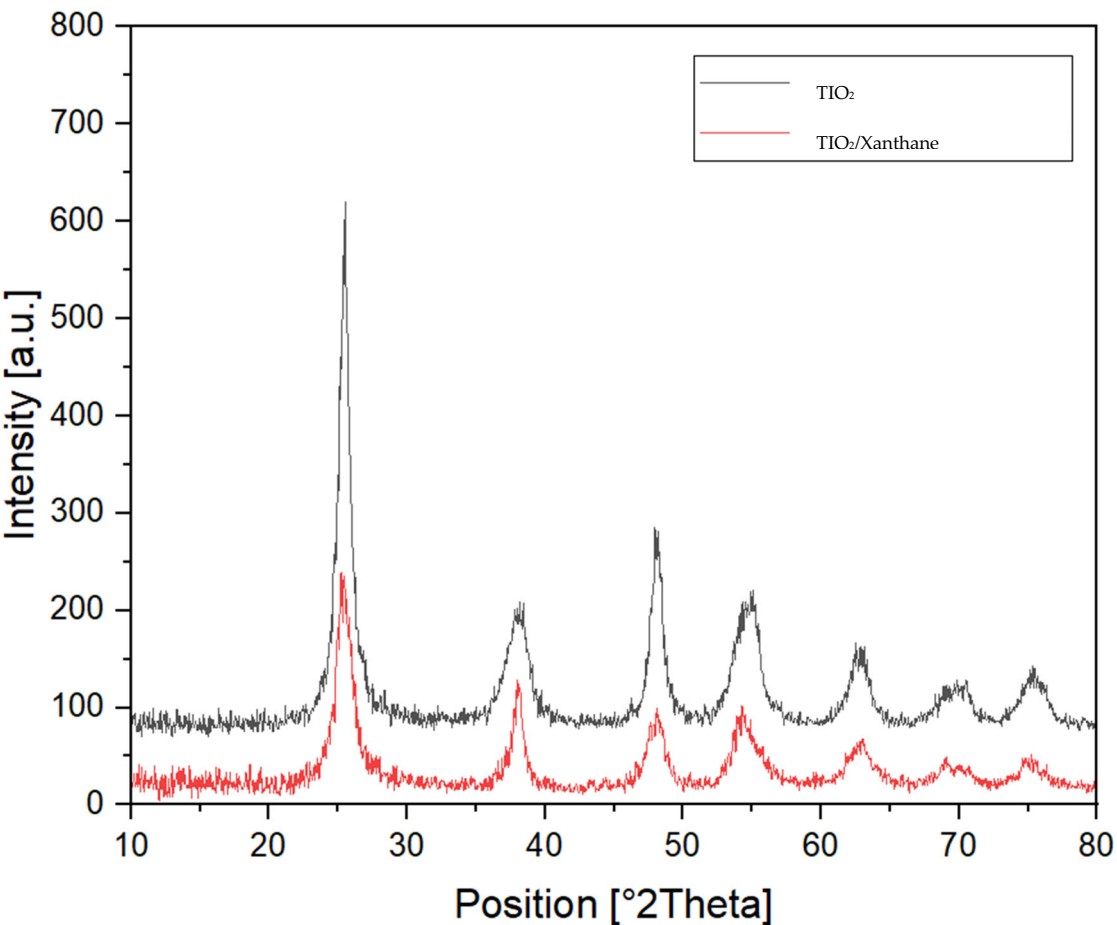

**Figure 7.** XRD pattern of $TiO_2$/XG.

*3.2. Adsorption and Photocatalytic Activity of the $TiO_2$/XG Composites*

3.2.1. pH

The pH value is among the most essential factors influencing the elimination of pollutants due to its effect on the ionization of the pollutant and catalyst in the solution [36]. The results for the RR and RT dye removal efficiency via adsorption (first 30 min) and photodegradation using $TiO_2$/XG at different values of pH (3, 5, 7, and 11) at 25 mg/L of initial dye concentration, 25 mg/L of $TiO_2$/XG, and 400 mg/L of $H_2O_2$, are shown in Figure 8. According to the results, a slight degradation was observed in the darkness and increased by increasing irradiation time. This figure also shows that the degradation efficiency increased by increasing the pH value from 3 until it reached 5, then further increase decreased the removal percentage of RR and RT dye, respectively. The highest RR and RT dye removal efficiencies after 30 min were 31 and 42.2%, respectively, at pH 5. At an irradiation time of 120 min, the maximum removal efficiency was achieved at 98% and 92.8% for the RR and RT dyes, respectively. pH level change influenced the ionization degree and superficial charge of the adsorbent as well as the electrostatic interactions of the surface catalyst and the separation of functional groups on the catalyst's active sites, altering the solution's chemistry [37].

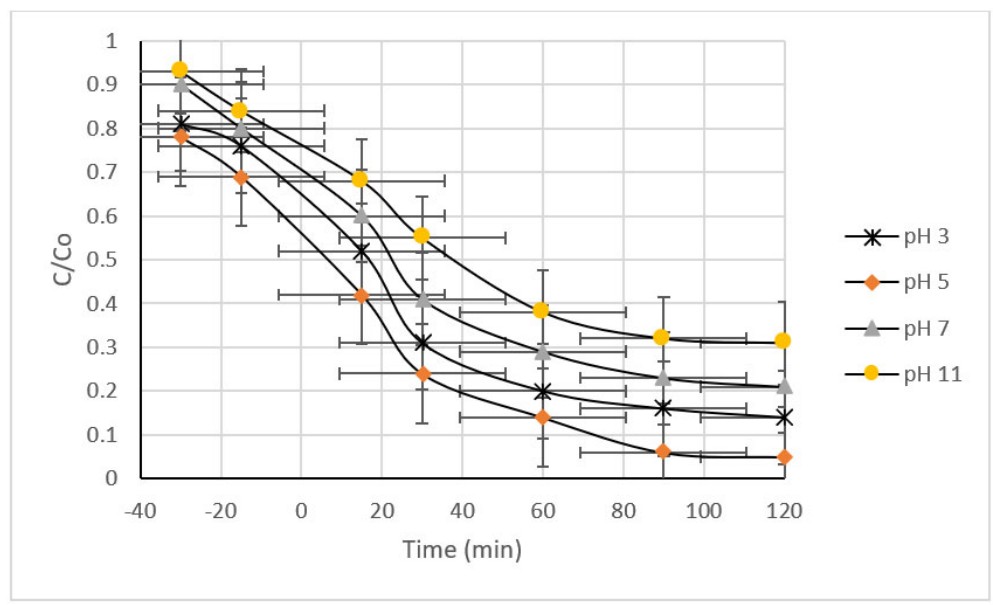

(**a**)

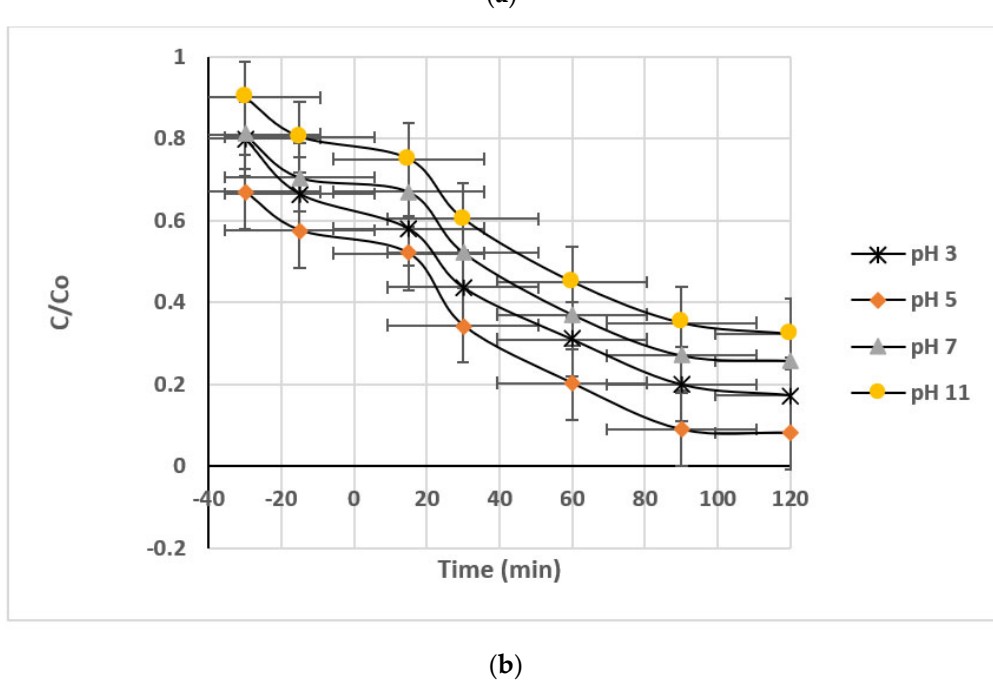

(**b**)

**Figure 8.** RR and RT dyes' removal percentage as a function of pH in the photocatalytic system. (**a**) RR dye, (**b**) RT dye.

In acidic solutions (pH < 5), the photodegradation of the dye was retarded by the high concentration of protons, resulting in lower degradation efficiency. In an alkaline medium (pH > 10), on the other hand, the presence of hydroxyl ions neutralizes the acidic end-products that are produced by the photodegradation reaction [38]. This is associated with the surface's ionization state, in line with the reactions below (Equations (4) and (5)), and in addition to the state of reactant dyes and acids and amines, among other products:

$$TiOH + H^+ \leftrightarrow TiOH_2{}^+ \tag{4}$$

$$TiOH + OH \leftrightarrow OH^- \leftrightarrow TiO + H_2O \tag{5}$$

A sudden drop in degradation was observed when the initial pH of the reaction mixture was kept at 11. The reactive blue dye substituents of an electron-donating group, such as—NH$_2$ in the $\alpha$-positions of the carbonyl group form intramolecular hydrogen bonds at high pH values [38]. Hence, the difference in the pH values can influence dye molecule adsorption onto the TiO$_2$ surface, which is a crucial step in photocatalytic oxidation [28].

The pH zero-point charge (pH$_{zpc}$) of TiO$_2$/XG was determined using the method described in Mohseni-Bandpi et al. [39]. This conclusion was important in determining the surface charge of TiO$_2$/XG against the solution, and their results are presented in Figure 9. The graph shows an intersection at 8.1 and 8.3 for the RT and RR dyes, respectively, demonstrating that at this pH level, there is a negative surface above TiO$_2$/XG and a positive surface below.

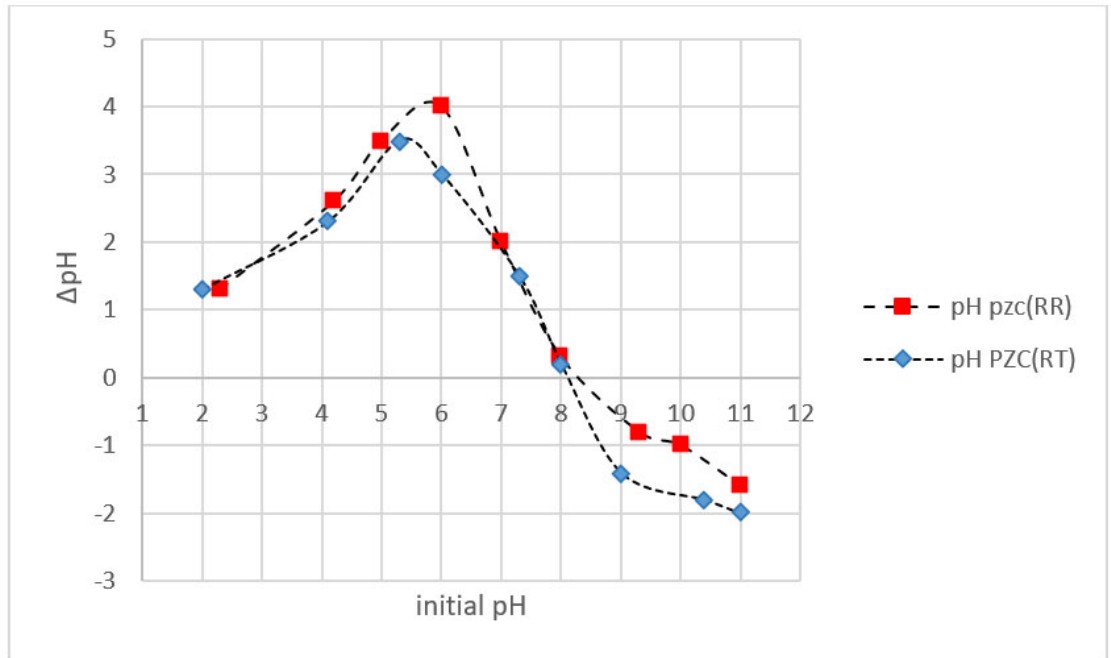

**Figure 9.** pH$_{pzc}$ for TiO$_2$/XG.

3.2.2. Dye Concentration and Time

To examine the effectiveness of TiO$_2$/XG for RR and RT dye removal at different initial dye concentrations (25, 50, and 75), experiments were carried out at pH 5, 25 mg/L of TiO$_2$/XG, a H$_2$O$_2$ concentration of 400 mg/L, and a 240 min reaction time. The results are presented in Figure 10. The decolorization efficiency declined as the initial RR and RT dye concentrations increased; this may have been because the dye molecules were increasingly adsorbed onto the surface of TiO$_2$ with the increase in the initial dye concentration. Because there is no direct contact, it is possible that the significant dye adsorption inhibited the interaction between the dye molecules and the hydroxyl radicals or photogenerated holes. With more dye, light is absorbed by the dye molecules, meaning no photons can reach the surface of the photocatalyst, reducing the photocatalytic decolorization efficiency [40]. In addition, prolonging the time by up to 90 min increased the dye degradation, with maximum removal of 94 and 91 for the RR and RT dyes, respectively. The quantity of absorbed photons on the catalyst surface increases as the radiation time increases, promoting the photocatalytic process [41]. Again, increasing the catalyst concentration to an optimum level improved the degradation of the dye. Following that, the elimination percentage decreased. These results are in agreement with the findings of Inamuddin [24].

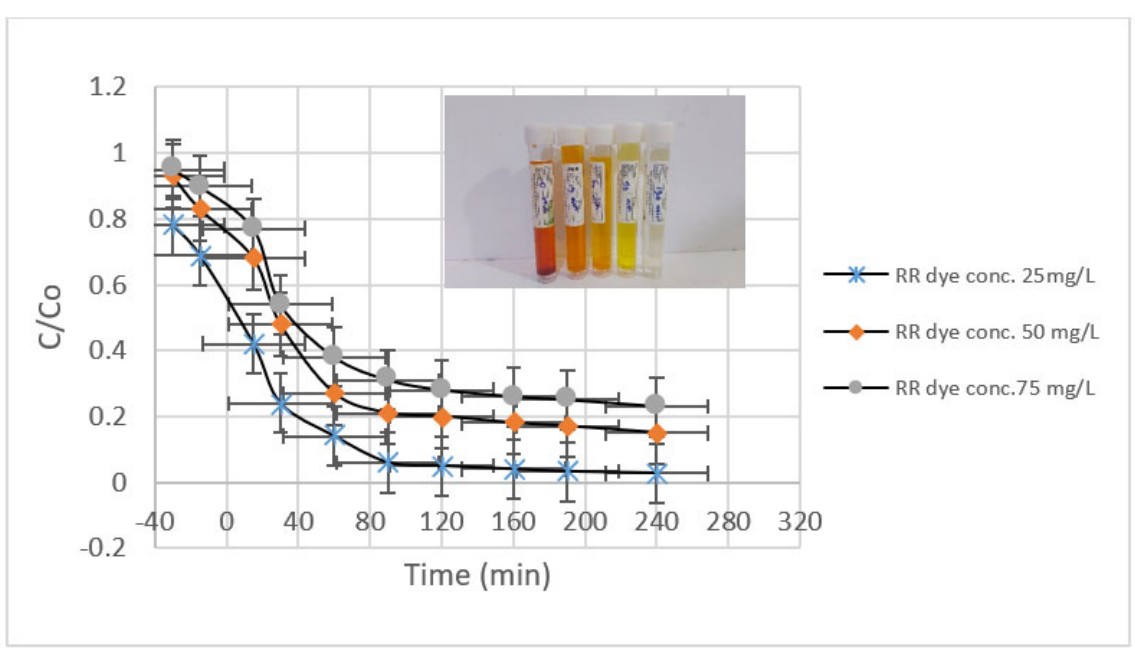

(**a**)

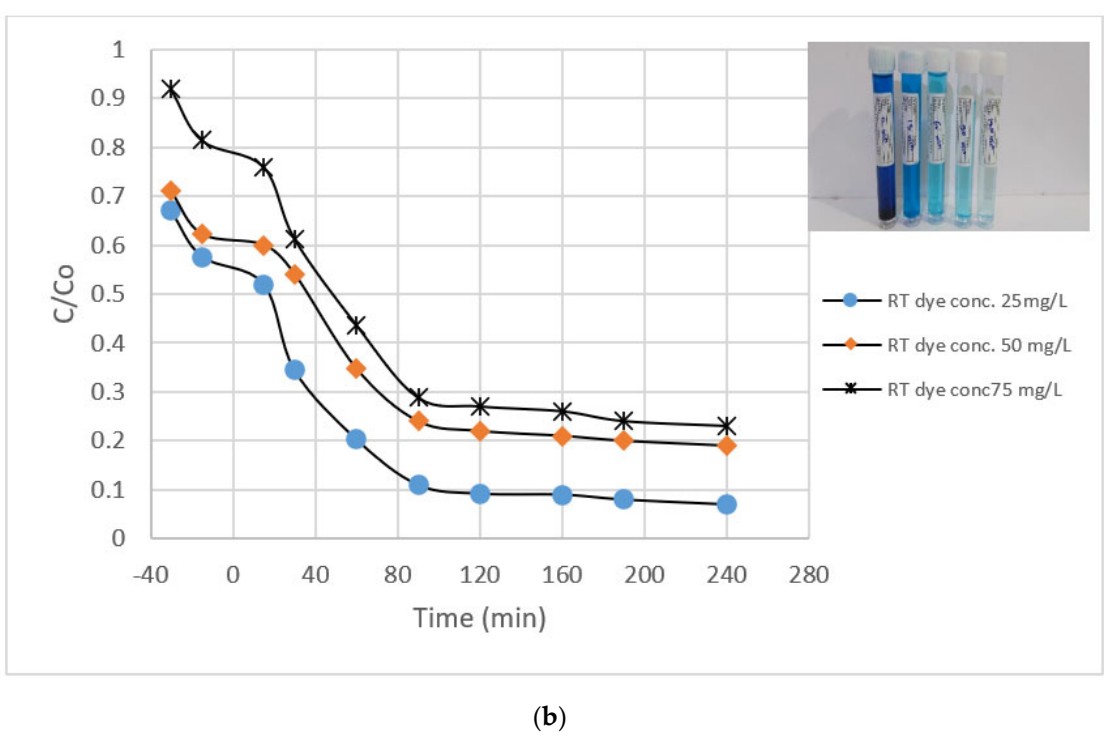

(**b**)

**Figure 10.** Dye degradation as a function of their concentration and time. (**a**) RR dye, (**b**) RT dye.

### 3.2.3. Effect of TiO$_2$/XG

A series of experiments were carried out in which TiO$_2$/XG concentrations (15, 25, 50, and 75) mg/L were varied while the remaining parameters remained constant. As displayed in Figure 11, adsorption data can be seen for reaction times of −30 to 0 min, in line with the 0–30 min adsorption time interval. Meanwhile, the 0–120 reaction time range represents the photodegradation data. Increasing the catalyst concentration from 15 to 25 mg/L led to a further increase in the RR and RT dye adsorption removal and photodegradation removal efficiencies. A further increase in the amount of TiO$_2$/XG

nanocomposite above 25 mg/L decreased the degradation rate. Additional TiO$_2$/XG additions did not affect the degradation process and they merely widened the layer formed in the solution [24].

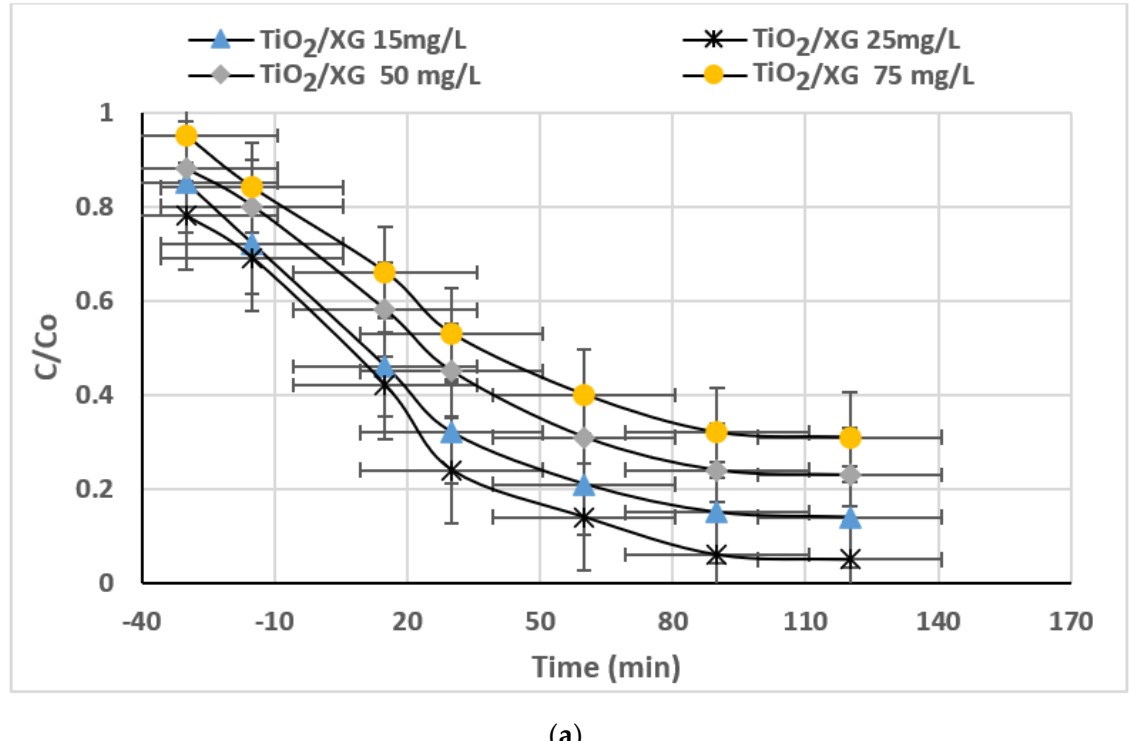

(**a**)

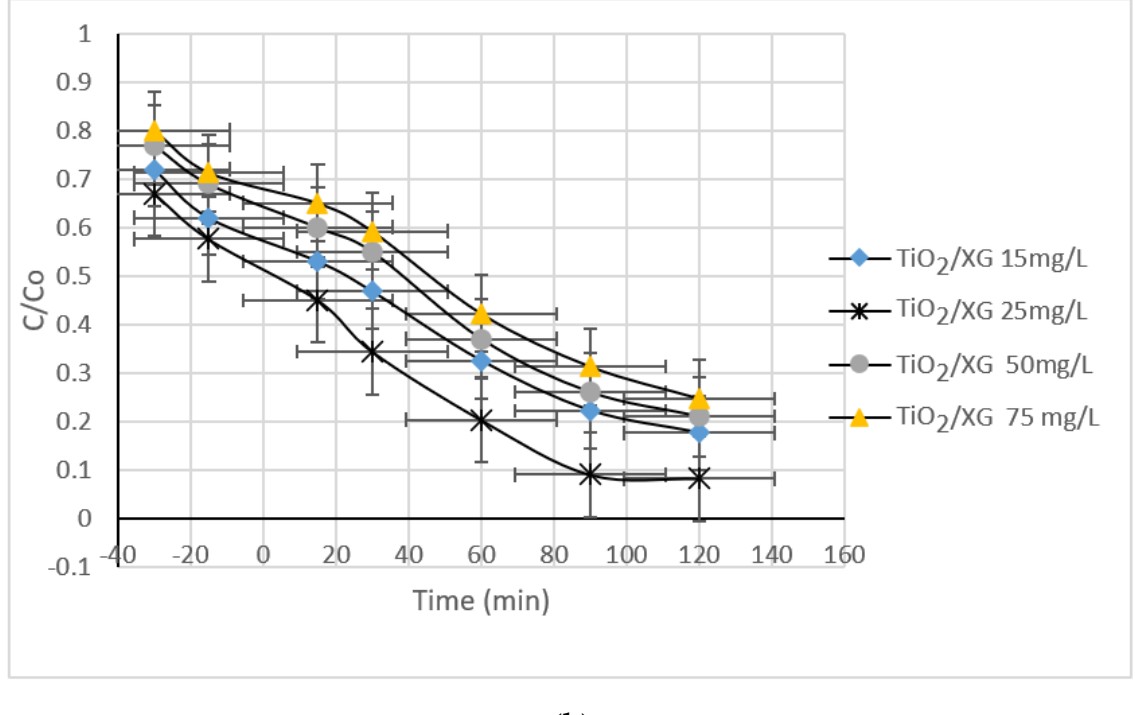

(**b**)

**Figure 11.** Effect of TiO$_2$/XG concentration on the degradation of (**a**) RR dye and (**b**) RT dye degradation.

### 3.2.4. $H_2O_2$ Concentration

The $H_2O_2$ concentration is a significant factor in the degradation of dyes as it is the source of hydroxyl radicals. Figure 12 shows the variation in the RR and RT dye degradation from an aqueous solution at different $H_2O_2$ concentrations (200, 400, and 600 mg/L). It can be noticed from this figure that increasing the initial concentration of $H_2O_2$ from 200 to 400 mg/L increased the decolorization efficiency from 84 and 82.3% to 92.5 and 90.8% for RR and RT dyes, respectively, in line with Equations (6)–(8). Further increases in $H_2O_2$ concentration to 600 mg/L would reduce the decolorization efficiency, whereby high concentrations of $H_2O_2$ act as a radical scavenger according to Equations (9) and (10), and $HO_2^\bullet$ becomes predominant. Reduced concentration of $H_2O_2$ reduces the hydroxyl radicals ($OH^\bullet$), leading to insufficient dye consumption and a reduction of the oxidation rate; this was also noted by [42,43].

$$TiO_2(e^-) + H_2O_2 \rightarrow TiO_2 + OH^- + {}^\bullet OH \tag{6}$$

$${}^\bullet O^{-2} + H_2O_2 \rightarrow OH^- + {}^\bullet OH + O_2 \tag{7}$$

$$H_2O_2 + hv \rightarrow 2 {}^\bullet OH \tag{8}$$

$$H_2O_2 + {}^\bullet OH \rightarrow HO_2^\bullet + H_2O \tag{9}$$

$$HO_2^\bullet + {}^\bullet OH \rightarrow H_2O + O_2 \tag{10}$$

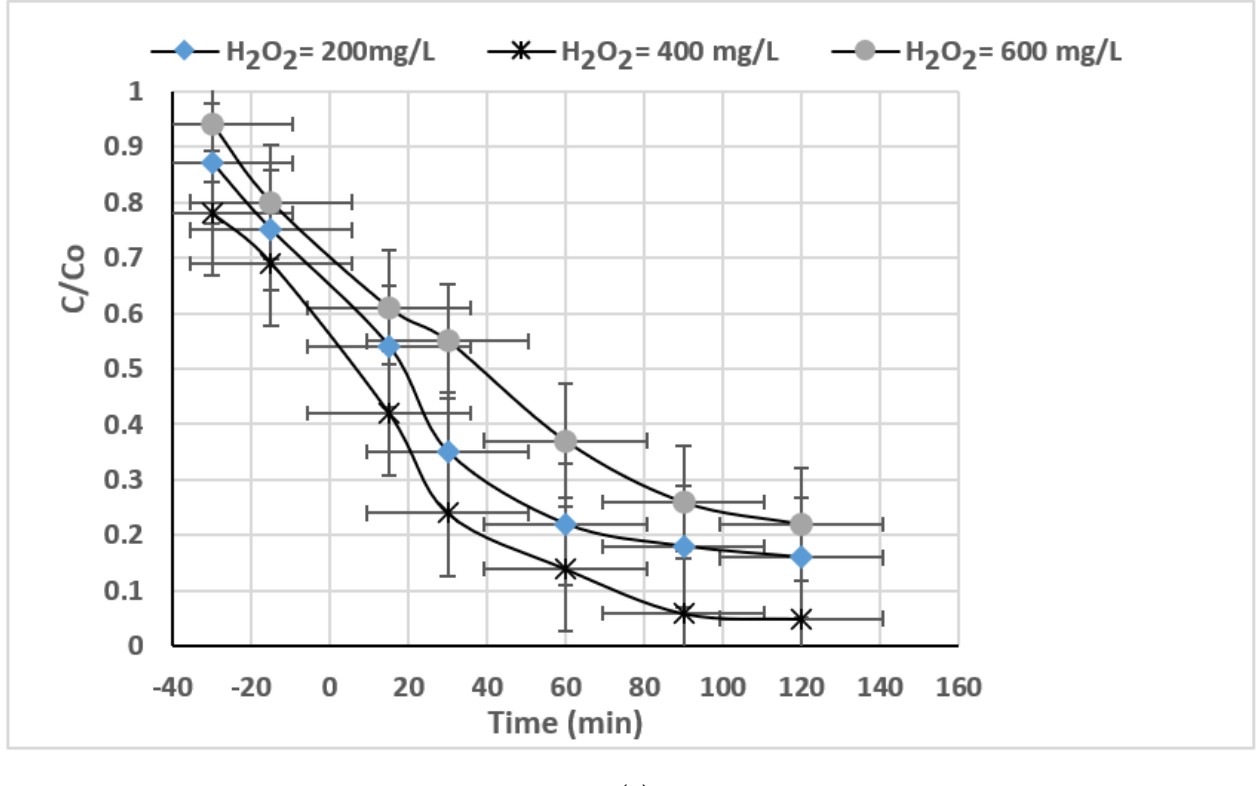

(a)

**Figure 12.** *Cont.*

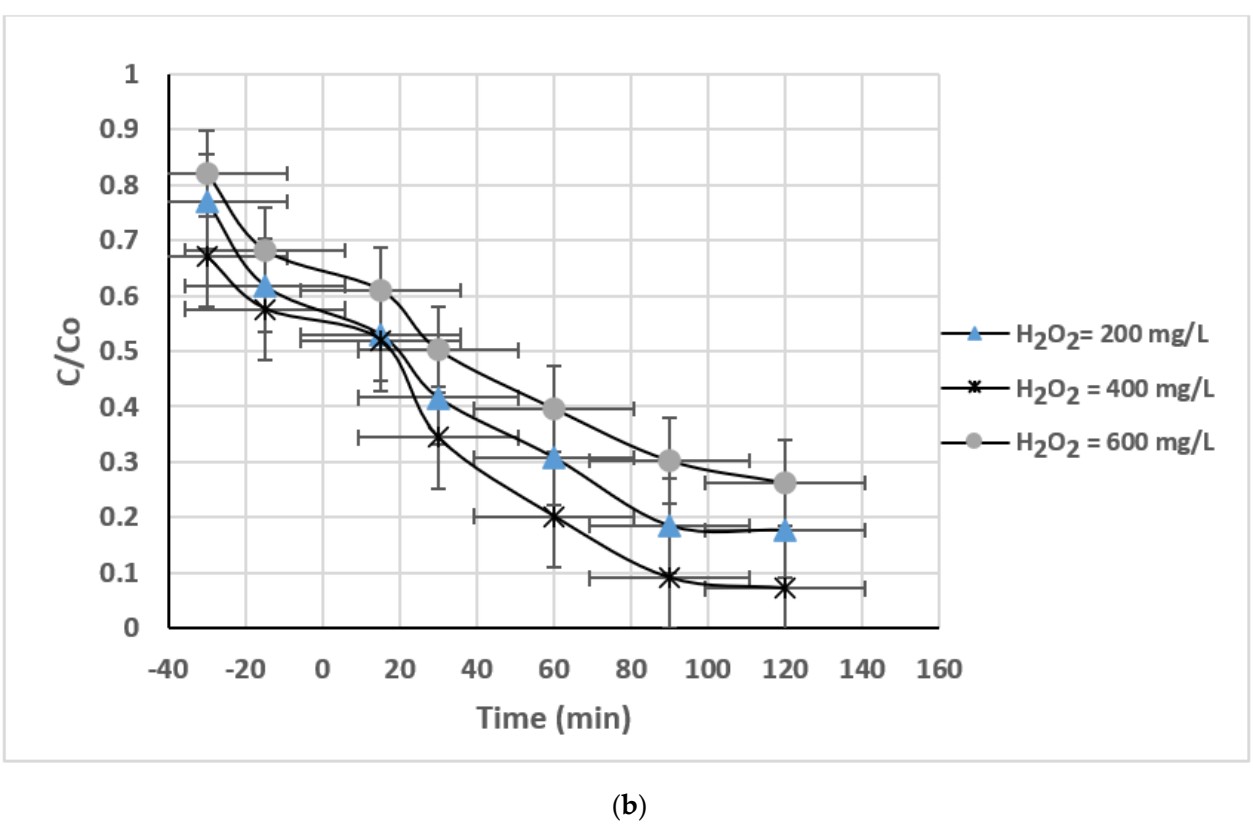

(**b**)

**Figure 12.** The influence of $H_2O_2$ concentration on the degradation of (**a**) RR dye and (**b**) RT dye.

### 3.3. Photodegradation Kinetics

To examine the performance of solar photocatalytic processes for the degradation of RR and RT dyes from aqueous solution, kinetic analysis is essential for understanding the photodegradation process and reaction rate, which depend on the dynamic collaboration of dyes and the nanocomposite surface. A pseudo-first-order kinetics model is often used to show the reaction rate of a heterogeneous catalytic process, which can also be explained via the Langmuir–Hinshelwood (L–H) kinetic model [44–46].

$$\ln \frac{C_o}{C} = k_{obs} \cdot t \tag{11}$$

where $K_{obs}$ is the observed pseudo-first-order reaction rate constant (1/min), $C$ and $C_o$ are, respectively, the dye concentrations (mg/L) following exposure time $t$ and the dye concentrations at the start (mg/L), and $t$ is the exposure time (min).

Plotting $\ln\left(\frac{C_o}{C_t}\right)$ as a function of t from 0 to 120 min produces the kinetics results, as shown in Figure 13, and Table 2 gives the results for $K_{obs}$ values as well as the regression coefficients ($R^2$) of the linear plot equations. It can be seen that increasing the initial concentrations of the RR and RT dyes reduced the results for the constant degradation rate of $K_{obs}$. The finding from the kinetics model show that the kinetics pseudo-first-order model is an appropriate model to describe the reaction rate, and for all different concentrations, this model properly covered the results. Thus, the values of the explanation coefficients ($R^2 > 90\%$), and the reaction rate is reduced by increasing the RR and RT dye concentration. At high concentrations, due to the increased concentration of intermediate products, the number of active hydroxyl radicals was limited, and as a result, the constant degradation rate was decreased. This agrees with the experimental findings in Section 3.2.2 on how the initial RR and RT dye concentrations influence the dye removal efficiency.

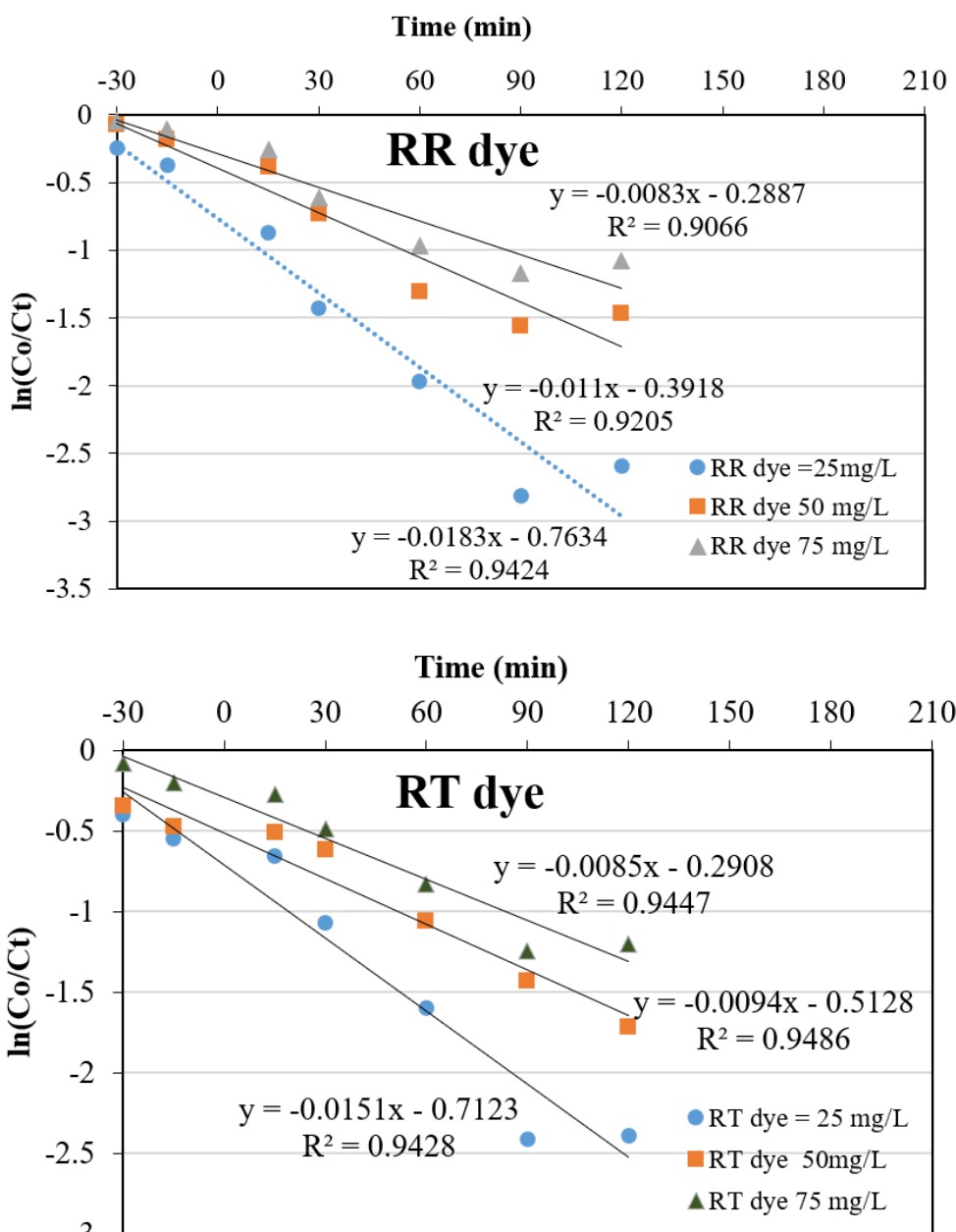

**Figure 13.** The kinetic curves pseudo-first-order equation for RR dye and RT dye degradation at different initial concentrations.

**Table 2.** Reaction rate parameters for dyes decomposition in the solar photocatalytic process.

| Dye Concentration | First-Order for RR Dye | | | First-Order For RT Dye | | |
|---|---|---|---|---|---|---|
| | Removal % at 120 min | $R^2$ | $K_{obs}$ (min$^{-1}$) | Removal % at 120 min | $R^2$ | $K_{obs}$ (min$^{-1}$) |
| 25 | 92.5 | 0.9424 | 0.0183 | 90.8 | 0.9428 | 0.0151 |
| 50 | 77 | 0.9205 | 0.011 | 74 | 0.9486 | 0.0094 |
| 75 | 66 | 0.9066 | 0.0083 | 70 | 0.9447 | 0.0085 |

## 4. Conclusions

In the present study, TiO$_2$ immobilized on xanthan gum (denoted as TiO$_2$/XG), prepared using the sol–gel dip-coating technique, was tested for the photocatalytic removal of RR and RT dyes under solar light, as identified using FTIR, XRD, SEM, and UV–Vis tech-

niques. The best solar photocatalytic degradation rates, namely 92.5 and 90.8% at 25 mg/L of RR dye and RT dye, respectively, were achieved at pH 5 after an irradiation time of 120 min with 25 mg/L of $TiO_2$/XG and 400 mg/L of $H_2O_2$. The application of immobilized $TiO_2$ thus facilitates the recovery of materials while avoiding secondary pollutant problems involving the escape of a photo-catalyst into a water medium.

The results presented in this study confirm that the $TiO_2$ particles enhance the mechanical and catalytic properties of XG and, thus, offers good potential for applications as a catalyst for degrading and eliminating organic pollutants of pharmaceutical origin that persist in the aqueous environment.

**Author Contributions:** A.I.A. and N.A.M.: editing, methodology, data analysis, and experiments. A.A.M.: methodology and supervision. T.J.A.-M.: writing, results and discussion, and editing. All authors have read and agreed to the published version of the manuscript.

**Funding:** This research received no external funding.

**Institutional Review Board Statement:** Not applicable.

**Informed Consent Statement:** Not applicable.

**Data Availability Statement:** The authors of this study confirm that the data supporting the findings of this study are available within the manuscript.

**Acknowledgments:** The authors are grateful for the financial and scientific support from the University of Baghdad and Al-Mustaqbal University.

**Conflicts of Interest:** The authors declare that they have no known competing financial interests or personal relationships that could have appeared to influence the work reported in this manuscript. The authors of this study declare no competing interests.

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
