# Peer review of "Solar-Induced Photocatalytic Degradation of Reactive Red and Turquoise Dyes Using a Titanium Oxide/Xanthan Gum Composite"

_sustainability, doi:10.3390/su151410815_

Round 1

Reviewer 1 Report

1. Suggest the author to use full form for the material they used in the title.

2. Abstract: Please list the adsorption is done in batch process or continuous process. What type of first-order kinetics model? This the proposed material acting as a catalyst or absorbent? If it is a catalyst, what reaction it has undergone?

3. Materials and Methods: Kindly state the country of the materials. How the authors ensure the purity of RR and RT under used conditions or to define the concentration of these used RR and RT dyes?

4. Materials and Methods: Missing TiO2/ XG Characteristics procedures on SEM etc.

5. Section 3.1.1 and section 3.3: There are steps should be placed in section 2, section 3 supposed to be results and discussion.

6. Section 3.1.2: All peaks should discuss on the potential rotation, stretching on different bonds or functional groups, such as 1000 and 1500 cm-1. What caused the reduction of 1500 cm-1in Tercouse and what caused the reduction of 1000 cm-1in red colour?

7. Section 3.3: After the finding of the first-order kinetics model, what is the finding from here?

8. What reaction(s) involved here? Oxidation? Absorption? Or combination of different reactions?

9. Kindly proof read the manuscript, for example, conclusion: XRD- SEM, it should be XRD, SEM.

10. Fig 3 is missing magnetic stirrer bar.

11. Kindly use the same size for the Fig 6 (a), right hand size, 100nm not 200nm to standardize the format.

12. Insert error bar for Fig 8, 9, 10, 11 and 12.

13. Smoothen Fig 7.

Author Response

The reviewer’s comments are highly appreciated as these have considerably improved our manuscript. We look forward to your positive response.

Reviewer 2 Report

1. Usually in such studies no aid except the catalyst is used. Why you used H2O2. Please explain the mechanism of degradation with a scheme taking into account H2O2.  

2. Why at PH 3 the degradation is decreasing. Explain it, keeping in mind the mechanism of degradation.

3.  Add the UV plots of degradation to the manuscript or as supplementary data .

4. Add the digital images of the dyes concentration before and after degradation.

5. Did you calculated the particles size of the TIO2. Particles size and surface area will have an effect on the degradation. 

6. If possible, please also carry out BET for surface area measurement

Author Response

The reviewer’s comments are highly appreciated as these have considerably improved our manuscript. We look forward to your positive response

Reviewer 3 Report

Comments:

·       In this research work (sustainability-2134923; Solar-induced Photocatalytic Degradation of Reactive Red and Turquoise Dyes using a TiO2/XG Composite), authors explores the solar-induced photocatalytic degradation of reactive red and reactive turquoise dyes in a single system using TiO2 immobilized in xanthan gum (TiO2/XG), synthesized using the sol-gel dip-coating technique for direct precipitation. SEM-EDX, XRD, FTIR, and UV-Vis were used to assess the characteristics of the resulting catalyst. Moreover, the effects of different operating parameters, specifically pH, dye concentration, TiO2/XG concentration, H2O2 concentration, and contact time, were also investigated. I recommend the authors should address the following "minor Issues" and add some more data before the paper is accepted for publication in "sustainability " journal.

·       In line # 37, “organic compounds, dyes, and other nutrients, which, if not treated properly,” what type of nutrients/dyes. Authors need to add some examples so that the readers can understand it easily. Authors can take help from; FE-ZRO2 imbedded graphene like carbon nitride for acarbose (ACB) photo-degradation intermediate study. Carbon nanotubes heterojunction with graphene like carbon nitride for the enhancement of electrochemical and photocatalytic activity. Fabrication of spherical-graphitic carbon nitride via hydrothermal method for enhanced photo-degradation ability towards antibiotic

·       The authors just mentioned the company name from where they took the RR and RT. Need to provide more details, the % percentage of impurity, and the CAS number, if possible, so the researchers can easily repeat this experiment for their comparative study. Must add this information.

·       Line #184, “TiO2/XG prior to the treatment, with a semi-spherical morphology”. Authors claim the semi-spherical morphology, while according to the SEM images (figure 6) it does not look semi-spherical, it looks like a cluster of a particle with irregular morphology. Authors need to explain morphology data accordingly.

·       The authors need to add the XRD pattern of TiO2 in Figure 7 with TiO2/XG, so the readers can easily understand it and compare it very well.

·       Authors mentioned in line # 207 that “TiO2/XG at different values of pH (3, 5, 7, and 11)” how authors changed the pH values ? by varying the dye concentration or TiO2/XG .? did author measure the photocatalytic activity of solo TiO2 .? Author can take help from: Under vacuum synthesis of type-I heterojunction between red phosphorus and graphene like carbon nitride with enhanced catalytic, electrochemical and charge separation ability for photodegradation of an acute toxicity category-III compound. rection of duct-like graphitic carbon nitride with enhanced photocatalytic activity for ACB photodegradation.

·       In line # 238; concentrations (25, 50, and 75), what is the unit of these concentrations? Authors need to carefully revise the manuscript and remove these type of typographical mistakes.

·       The authors used both dyes (RR, RT), but in colored pollutants, the photo-degradation may also be due to the sensitization effect. ? did the authors check this ? Authors can check the synthesized catalyst with non-colored also. So the degradation ability can be verified.

·       The collusion part needs to be rewritten, as it looks similar to the abstract part. in the conclusion part, the authors just need to mention the final results output and their future aspects instead of mentioning the analytical techniques names/instruments names (line #318).

·       Also, if possible, try to improve the graphical representation; instead of making it complex, make them in generalized form so the readers can understand it on their first look. Such as; figure -2 (Schematic of the TiO2/XG preparation).

Author Response

(The authors gave the same response as above.)

Reviewer 4 Report

Dear Authors,

 Indeed, your work did not show significant novelty: use of TiO2 composite for dye degradation). In addition, the experimental work is relatively poor. Although the work is interesting, the manuscript cannot be published in the present form. Hereafter are some comments and questions, thus helping to revise your manuscript: the materials are not adequately characterized, the photochemical processes are poorly discussed, and the degradation pathway is not commented. Also, many parameters have not been assessed. 

Author Response

Dear Reviewer, thank you for the time you have spent reviewing our manuscript and for your positive consideration. The required corrections were made.

Reviewer 5 Report

The article present some novel results but should be improved before publication:

1. English style and grammar should be improved. Many typos are found

2. Typos are found in the text of figure 3.

3. The scale bars in the SEM images are not clear. Make them much bigger please.

4. The values of C/Co increases after 90 min of reaction, why? this is not normal. The values of C/Co should decrease with the time.

5. Authors should make scavenger experiments to find the oxydizing agents responsible of the dye degradation.

6. The introduction section is very short. Authors should add other recent works about degradation of dyes. The following references must be added to improve the background:

a) Journal of Environmental Management, Volume 315, 1 August 2022, 115204, https://doi.org/10.1016/j.jenvman.2022.115204

b) Environmental Science and Pollution Research volume 29pages76752–76765 (2022). https://link.springer.com/article/10.1007/s11356-022-21301-y

c) Topics in Catalysisvolume 65pages1102–1112 (2022), https://link.springer.com/article/10.1007/s11244-022-01690-7

7. The abstract must be re-written to emphasize the findings of this work. It looks like a description. relevant data must be mentioned.

8. The contribution of this work should be emphasized in the introduction section. it is not clear the advantages of this work with respecto to previous literature.

Author Response

Dear Reviewer, thank you for the time you have spent reviewing our manuscript and for your positive consideration. All the corrections were done.

Reviewer 6 Report

The manuscript (sustainability-2134923) describes the TiO2/XG Composite for photocatalytic degradation of reactive Red and turquoise dyes and the related studies. This article is interesting and well-studied, considering the significance of such research in recent times. TiO2 is a well-studied photocatalyst, and a few points in this manuscript should be clarified before we can accept the same for publication.  

 1.      In the introduction, the author should include and discuss the performance of a few other Xanthan gum-based photocatalysts or even other TiO2/XG composite photocatalysts reported earlier.

2.      Authors should provide UV-vis absorption spectra of the as-prepared materials.  

3.      How can be the practical water treatment set up based on the TiO2/XG composite photocatalyst?

4.      The mechanism of the photocatalysis process should be discussed with a suitable diagram.

5.      Minor issue: In figure 4, the captions are not complete and clear.

Author Response

(The authors gave the same response as above.)

Round 2

Reviewer 1 Report

All comments are addressed

Reviewer 2 Report

accept

Reviewer 4 Report

X

Reviewer 5 Report

The article can be accepted for publication.

Reviewer 6 Report

This manuscript can be accepted in its present form.